# Evaluation of the Effectiveness of a Bilingual Nutrition Education Program in Partnership with a Mobile Health Unit

**DOI:** 10.3390/nu16050618

**Published:** 2024-02-23

**Authors:** Madeleine L. French, Joshua T. Christensen, Paul A. Estabrooks, Alexandra M. Hernandez, Julie M. Metos, Robin L. Marcus, Alistair Thorpe, Theresa E. Dvorak, Kristine C. Jordan

**Affiliations:** 1Department of Nutrition and Integrative Physiology, University of Utah, Salt Lake City, UT 84112, USA; julie.metos@hsc.utah.edu (J.M.M.); t.dvorak@utah.edu (T.E.D.); kristine.jordan@hsc.utah.edu (K.C.J.); 2Department of Population Health Sciences, Spencer Fox Eccles School of Medicine, University of Utah, Salt Lake City, UT 84112, USA; joshua.christensen@hsc.utah.edu (J.T.C.); alistair.thorpe@hsc.utah.edu (A.T.); 3Department of Health and Kinesiology, University of Utah, Salt Lake City, UT 84112, USA; paul.estabrooks@health.utah.edu; 4Osher Center for Integrative Health, University of Utah Health, Salt Lake City, UT 84112, USA; alex.m.hernandez@utah.edu; 5Department of Physical Therapy and Athletic Training, University of Utah, Salt Lake City, UT 84108, USA; robin.marcus@hci.utah.edu

**Keywords:** nutrition education, dietary behavior, healthy eating, culinary instruction, health coaching, teaching kitchen, RE-AIM framework, social determinants of health

## Abstract

There are limited reports of community-based nutrition education with culinary instruction that measure biomarkers, particularly in low-income and underrepresented minority populations. Teaching kitchens have been proposed as a strategy to address social determinants of health, combining nutrition education, culinary demonstration, and skill building. The purpose of this paper is to report on the development, implementation, and evaluation of Journey to Health, a program designed for community implementation using the RE-AIM planning and evaluation framework. Reach and effectiveness were the primary outcomes. Regarding reach, 507 individuals registered for the program, 310 participants attended at least one nutrition class, 110 participants completed at least two biometric screens, and 96 participants attended at least two health coaching appointments. Participants who engaged in Journey to Health realized significant improvements in body mass index, blood pressure, and triglycerides. For higher risk participants, we additionally saw significant improvements in total and LDL cholesterol. Regarding dietary intake, we observed a significant increase in cups of fruit and a decrease in sugar sweetened beverages consumed per day. Our findings suggest that Journey to Health may improve selected biometrics and health behaviors in low-income and underrepresented minority participants.

## 1. Introduction

Chronic diseases, including diabetes and cardiovascular disease, are widespread [1,2,3] and costly for patients and healthcare systems alike [4,5]. Approximately 60% of U.S. adults have at least one chronic disease, while 40% have two or more [6]. Annually, chronic diseases in the U.S. translate to trillions of dollars in health care expenditures and lost productivity [7]. The underlying mechanisms of diabetes and cardiovascular disease can be linked in part to a excessive consumption of sodium, saturated fat, and added sugar, and insufficient consumption of fruits, vegetables, whole grains, and fat-free or low-fat milk products [8,9,10]. This dietary imbalance is problematic, as fruit and vegetable consumption is inversely associated with chronic disease risk [11,12] and mortality [13,14]. Adding to this problem is the context that the cascading impact of dietary health behaviors and chronic disease is not evenly distributed amongst the U.S. population and individuals belonging to historically marginalized groups [1,2,3] or those with low socioeconomic status [15] experience disproportionately higher rates of diet-related chronic disease. These disparities are the result of a range of social determinants of health [16,17] that include differential access to evidence-based health promotion and disease prevention interventions [18] and nutrient dense foods [19].

The magnitude and severity of nutrition related chronic diseases in the U.S. warrants creative interventions that address modifiable risk factors. Preliminary lifestyle intervention programs were primarily didactic, utilizing knowledge acquisition as the main mechanism to promote behavioral change [20,21]. The National Diabetes Prevention Program (DPP), established by the Centers for Disease Control and Prevention, takes a goal-based behavioral approach, providing both didactic and skill-based learning experiences based on a randomized, controlled clinical trial [22,23,24]. Modified group versions of the DPP improved the cost-effectiveness and availability of the intervention [25]; however, the program requires a minimum of 22 classes within 12 months to receive CDC-recognition [26], which can be impractical for participants already facing significant social determinants of health, and sponsoring organizations alike [27,28,29]. A review of translational studies examining the U.S. and Finnish DPP concurred that less intensive interventions could increase feasibility without losing meaningful impact on biomarkers [30].

Many newer community-based lifestyle interventions have shifted towards a skill acquisition framework, which emphasizes hands-on learning to bolster self-efficacy for health behavior change [31,32,33]. Examples of these programs include culinary medicine and teaching kitchens in university, hospital, remote, and community settings. Teaching kitchens and other nutrition education strategies delivered alongside culinary demonstrations have been proposed as a strategy to address social determinants of health and related health inequities by combining nutrition education, skill building approaches, and the provision of healthy food through intervention sessions or food prescriptions [34,35,36].

Examples of successful nutrition programs that involve culinary education include the Cooking Matters for Adults intervention, which demonstrated efficacy in improving food resource management skills for shopping and cooking healthy meals, as well as self-confidence in low-income participants [37]. Additionally, a produce prescription and cooking education pilot program at a Federally Qualified Health Center showed improvements in cooking self-efficacy, behavior change strategies for healthy eating, and vegetable intake in racially and ethnically diverse adult participants [38]. Further, a systematic review investigating the impact of culinary interventions on psychosocial outcomes reported that community-based cooking interventions positively influenced socialization, self-esteem, and quality of life in adults [39]. Finally, in a systematic review of interventions that provided cooking experiences for participants, the outcomes suggested that cooking demonstrations alone may not be sufficient to achieve behavioral or cardiometabolic changes—but when combined with nutrition education appeared to be superior [40]. Positive findings, as reported in these programs, likely explain the shift towards employment of a skill acquisition framework in nutrition programming.

Despite these promising nutrition education and skill building intervention outcomes, there is limited information on the potential for scale up and translation of these interventions in the communities that could most benefit [41,42,43]. The use of planning and evaluation frameworks that explicitly focus on understanding individual level reach and effectiveness of healthy eating and health promotion interventions while also considering the setting level factors that will influence intervention adoption, implementation, and maintenance have shown promise in increasing the generalizability and scale up of health behavior interventions [44]. Specifically, the Reach, Effectiveness, Adoption, Implementation, and Maintenance (RE-AIM) Framework has demonstrated utility in expanding the public health impact of health behavior interventions when used in planning to create strategies designed specifically for dissemination, equity, and sustainability [45,46]. Applying the RE-AIM Framework in intervention planning also focuses on place-based strategies that take interventions to the local settings where populations that experience health disparities reside [47]. Using a place-based approach has been proposed to promote health and prevent disease by (1) addressing barriers to access (e.g., travel, time, cost), (2) aligning with local priorities and leveraging local assets to deliver the intervention, and (3) fostering community collaboration and cohesion [48].

The purpose of this paper is to report on the development, implementation, and evaluation of Journey to Health, a program designed for community implementation using the RE-AIM planning and evaluation framework. Based on empirical nutrition literature, the Journey to Health nutrition education program borrows elements from successful behavior change interventions, such as health coaching, goal setting, and didactic nutrition education classes, as well as elements from teaching kitchen interventions, including culinary demonstrations.

The significance of this study lies in our program, which was designed for a broad reach by (1) integrating the intervention into an existing community resource, the University of Utah Wellness Bus (Wellness Bus), that provides biometric screening and health coaching services to low-income and underrepresented minority community residents and (2) filling a community gap for free, bilingual, and easily accessible nutrition education by offering the program in four centrally located spaces where local residents already gather. Journey to Health was designed for effectiveness by implementing teaching kitchen components, such as bilingual nutrition and physical activity education, culinary demonstrations, and the use of motivational interviewing strategies and health coaching techniques, to effectively promote behavior change [35].

To assess reach, we monitored the number of participants who (1) registered, (2) attended at least one nutrition class, (3) completed at least two biometric screens, (4) completed at least two health coaching appointments, and (5) completed at least half of the program activities, which included, at minimum, two nutrition classes, one health coaching appointment, and one biometric screen (i.e., to assess change).

To assess effectiveness, we analyzed pre- and post-changes in body mass index (BMI), diastolic and systolic blood pressure, hemoglobin A1c, glucose, total cholesterol, high- and low-density lipoproteins, and triglycerides. The secondary effectiveness analysis focused on pre- and post-changes in self-reported health behaviors, including physical activity, fruit intake, vegetable intake, sugar sweetened beverage intake, artificially sweetened beverage intake, and dietary self-efficacy.

## 2. Materials and Methods

Using the RE-AIM Framework, the Journey to Health study was designed to create a nutrition education and skill building intervention with the potential to have a broad public health impact and address health disparities. This design included planning an intervention that would (1) reach many low-income and minority community residents who could benefit from nutrition education and skill building, (2) effectively improve health behaviors and biometric outcomes, and (3) be adoptable, implementable, and sustainable in similar communities experiencing health inequities. To address reach, we partnered with the University of Utah Wellness Bus that is supported through a legislative mandate and philanthropic gift to provide health promotion and chronic disease prevention screening and intervention for approximately 1000 people per year in communities experiencing higher rates of health disparities. To address effectiveness, we used the underlying principles and components of promising Teaching Kitchen approaches. To address adoptability, implementation, and sustainability we designed the intervention to fit with the local community priorities resources (e.g., Wellness Bus services; place-based approach for delivery). We used all RE-AIM dimensions in the intervention design and planning process but focused our evaluation pragmatically on community prioritized outcomes of reach and effectiveness.

The Journey to Health program was designed to integrate with the services of the Wellness Bus. The Wellness Bus is a mobile health unit that provides free preventative health care to several underserved communities in Salt Lake County. The Wellness Bus sees approximately 1000 people annually of which 65% are un/underinsured and 55% are Hispanic/Latino. As such, Journey to Health was made available to the communities that the Wellness Bus was already serving on a weekly basis. The only inclusion/exclusion criterion was that the participant must have been 18 years of age or older. Place-based (i.e., the Wellness Bus brining the opportunity into the community) recruitment strategies included an invitation to participate during a typical Wellness Bus visit, a referral from a Wellness Bus employee, and advertisements on the Wellness Bus social media platforms. In addition, place-based strategies were also used, including flyers placed in local public libraries, recreation centers, and food pantries which directed participants to attend the Wellness Bus, text/call study staff, or to enroll through the Wellness Bus webpage.

The Journey to Health program was developed to meet the core evidence-based components described as foundational to Teaching Kitchens. These included (1) science-based nutrition recommendations, (2) culinary demonstrations to develop skills to prepare healthy and affordable meals, (3) the independent and synergistic importance of physical activity promotion, (4) behavioral strategies to support the selection, preparation, and intake of healthful foods, and (5) health coaching to support sustained behavior change. In addition, small incentives were provided to increase access to healthful foods for program participants. The resulting program (See Table 1) was operationalized within a six-month period as four one-hour nutrition classes, three culinary demonstrations (embedded within two of the nutrition classes), two biometric screens, two heath coaching appointments, and a culminating community meal. A study team, consisting of a registered dietitian from the University of Utah Center for Community Nutrition and nutrition graduate students, delivered the nutrition classes and culinary demonstrations. In addition, Spanish interpretation was provided at each class (41% of participants indicated Spanish as their preferred language). Health coaching appointments were completed using the existing process on the Wellness Bus, by a registered dietitian. The culminating experience was a community meal, which acted as an informal focus group where participants had the opportunity to provide feedback on the program while sharing a meal.

Following enrollment, participants were encouraged to complete an initial biometric screen on the Wellness Bus. When participants arrived at the Wellness Bus, consent was obtained via English or Spanish language electronic consent forms. All study activities were accessible in English and Spanish. In addition to Spanish language interpreters during nutrition classes, biometric screens and health coaching appointments at the Wellness Bus were facilitated by bilingual community health workers and registered dietitians. Following the intervention, participants were asked to complete follow-up biometric screening on the Wellness Bus.

We used a pragmatic approach to measuring outcomes. Specifically, existing screening and evaluation tools employed on the Wellness Bus were used to assess the reach and effectiveness of Journey to Health. This screening included height and weight (used to calculate body mass index), diastolic and systolic blood pressure, hemoglobin A1c, glucose, total cholesterol, high- and low-density lipoproteins, and triglycerides. In addition to screenings, the Wellness Bus provides free health coaching sessions to approximately 600 people per year. During these sessions, health behaviors were measured and discussed with a registered dietitian who provided health coaching on the Wellness Bus. We used pragmatic self-report measures, adapted from the harmonized patient reported data elements identified through expert panel review based on brevity, validity, and sensitivity to change [49]. Self-reported health measures included questions regarding physical activity, fruit intake, vegetable intake, sugar sweetened beverage intake, artificially sweetened beverage intake, and dietary self-efficacy. Table 2 contains the questions used for each of the self-reported health measures.

Reach was reported using descriptive statistics to provide a sense of the number of interested participants, the number that completed at least one nutrition class and two biometric screenings (this provided a pre-post for biometric outcomes), and the number that completed at least one nutrition class and two coaching appointments (this provided a pre-post for behavioral outcomes). Effectiveness was analyzed using pre-post data for participants that completed one nutrition class and two biometric screenings for the biometric outcomes and using pre-post data for participants that completed one nutrition class and two health coaching appointments for the behavioral outcomes. For these analyses, participants were considered eligible if they attended at least one nutrition class and completed either of the following: two biometric screenings within pre-defined time parameters or completed two health coaching appointments, also within time parameters determined a priori. See Table 3 for a description of program activities and corresponding time parameters.

Changes in outcomes over the course of 6 months were assessed using paired sample *t*-tests to compare mean scores. Pairwise correlations were also run to determine if there were any relationships between the number of nutrition classes attended, biometric screens conducted, or health coaching appointments with changes in the primary and secondary outcomes. Two subsamples were used in the analysis: first, the participants who completed at least two biometric screenings and one nutrition class. For this subsample, we also ran subgroup analyses for each variable that included only those at risk (e.g., A1c for only participants that were in the prediabetes range or higher) to examine the effect on biometric measures for those considered at risk according to their initial measurements. Second, to help examine the impact of the Journey to Health program on health behaviors, the participants who completed at least two health coaching appointments and one nutrition class were used to make up a separate subsample.

## 3. Results

Twenty-seven cohorts of the Journey to Health program were offered at Wellness Bus locations over the span of two years. Regarding reach, a total of 507 individuals registered for the program, with 310 attending at least one nutrition class. Out of the 310 participants who attended one nutrition class, 110 also completed at least two biometric screens and 96 attended at least two health coaching appointments within the time parameters described above. One hundred and forty individuals completed at least half of the outlined program by attending at least two nutrition classes, one biometric screen, and one health coaching appointment. One hundred and twenty-nine participants met the analysis eligibility criteria (i.e., had completed two biometric screenings and/or two health coaching appointments, paired with at least one nutrition course attended). The mean age for the sample was 51.5 (+/−15.3) years old, primarily female (72.9%), and largely identified as Hispanic or Latino (62%). Additional participant characteristics can be found in Table 4. On average, participants attended approximately three nutrition classes, three biometric screenings, and two health coaching appointments during the study period. When a participant had more than two eligible biometric screenings within the study period, the first eligible screening was used as the baseline measure, with the follow-up screening determined by the closest available screening to six months following the initial screening. This timeframe was determined to best reflect the program as outlined, with screenings occurring at the beginning and conclusion of the six-month program. The average duration between the eligible biometric screenings for analysis was approximately 4.37 months (133 days). Similarly, if multiple health coaching appointments occurred within the eligible timeframe, the first occurring appointment served as the initial measure, with the follow-up chosen as the closest to one month following the initial. Again, this was meant to reflect the schedule of the program, with health coaching appointments meant to occur one month apart, during months 3 and 4, respectively. The average duration between eligible health coaching appointments was approximately 8 weeks (55 days).

Paired sample *t*-tests revealed significant mean differences in BMI, systolic blood pressure, diastolic blood pressure, and triglycerides (see Table 5). For those who met all three analyses inclusion criteria, only diastolic blood pressure and triglycerides were seen to have significant decreases between the two timepoints. Exploratory analyses were also conducted to determine any potential gender (male vs. female) or ethnic (Hispanic or Latino vs. Non-Hispanic or Latino) differences between timepoints for each of the biometric variables. The only statistically significant difference was observed when comparing changes in triglyceride levels by ethnicity, with the Hispanic or Latino subgroup demonstrating a larger decrease in triglyceride levels over time (−64.6, *p* = 0.04), when compared to the non-Hispanic or Latino subgroup.

When testing for differences among those at risk (see Table 6), significant decreases were observed in BMI, systolic blood pressure, diastolic blood pressure, total cholesterol, LDL, and triglycerides.

When examining any differences in health behaviors between timepoints (see Table 7), we observed a significant increase in the cups of fruit consumed each day, as well as a decrease in the amount of sugar sweetened beverages. A significant increase in cups of fruit was also seen among those who met all three analyses inclusion criteria. Consistent with the biometric results, exploratory analyses were conducted to identify potential differences between gender and ethnicity on changes to self-reported health behaviors. No statistically significant differences were observed for any of the self-reported health behavior variables when stratified by gender or ethnicity.

The correlations between the number of nutrition classes attended, biometric screens conducted, or health coaching appointments revealed the only significant relationships occurred between the change in A1c and the number of biometric screens (r = −0.31, *p* = 0.01) as well as the change in artificially sweetened beverage consumption and the number of nutrition classes attended (r = 0.33, *p* = 0.01).

## 4. Discussion

Journey to Health was developed using the RE-AIM planning and evaluation framework with a goal to create a broad reaching intervention that was effective in supporting improvements in health and health behaviors while being accessible to community members who could benefit from nutrition education and skill building, but do not typically have that access. Using this approach, we developed an intervention that was integrated into existing community resources and showed promise for both reach and effectiveness. These findings led to 5 generalizations:(1) Journey to Health was attractive to the intended audience, but high initial interest did not result in high levels of intervention completion; (2) for those who engaged in Journey to Health, participation was related to improvements in health and health behaviors, especially those initially at higher risk; (3) there appeared to be a lack of relationship between the number of nutrition classes and health coaching appointments completed and health outcome changes contributing to the ongoing challenge of determining dose response relationships in health behavior intervention research; (4) integration of nutrition education with skill building strategies in existing community resources can provide a pathway to intervention sustainability; and (5) using processes that consider individual and setting level outcomes during the planning of interventions appears to be applicable to community-focused nutrition interventions.

To date, little is known about the reach of nutrition education interventions, particularly when focusing on groups that experience health disparities. Typically, studies that have focused on nutrition education and skill building have provided a description of the sample size that received the intervention with little information about the potential denominator from which it is derived [34,41]. Retention of participants is also seldom reported. In our project, based on the annual number of unique visitors to the Wellness Bus (~1000) and the two-year recruitment period, 25% of Wellness Bus clients expressed an interest in participating and 20% attended at least one nutrition class. While this proportion of Wellness Bus clients appears promising, it is necessary for others performing this type of community research to more systematically address reach to allow for comparisons across studies [48,50].

There are limited reports in the literature of community-based, nutrition education with culinary instruction that measure biomarkers, especially in low-income and underrepresented minority populations [34]. Our findings suggest that Journey to Health was able to bring nutrition education and skill building approaches to an underrepresented population and achieve significant change in outcomes. Our findings are inconsistent with a 2019 systematic review of culinary interventions that found no significant effect of the intervention on BMI, systolic blood pressure, diastolic blood pressure or LDL cholesterol [40]. We found that participants who had engaged in Journey to Health realized significant improvements in BMI, blood pressure, and triglycerides. Further, when considering higher risk participants, we also saw significant improvements in total and LDL cholesterol.

One potential explanation between our findings of effectiveness and the overall lack of effect of culinary interventions on metabolic outcomes could be the intensity of the intervention [40]. Our data would suggest that there may not be a potential dose response relationship between intervention intensity and improvement in outcomes. We found no relationship between changes in outcomes and the number of nutrition classes or health coaching appointments attended. However, when comparing Journey to Health outcomes to much more intensive interventions, it appears that the magnitude of effect is small in our sample. For example, Dasgupta et al. evaluated the impact of meal preparation training, physical activity, and nutrition education intervention in adults with type 2 diabetes mellitus (*n* = 72) in Canada, on body weight, glycemia, and blood pressure [51]. The intervention included 15 three-hour group education sessions at local grocery stores, for a 24-week duration, with group activities, including meal preparation under the supervision of a professional chef, walks, and nutrition education with a registered dietitian, as well as individual pedometer step recording. Results indicated biometric improvements in weight (mean change −2.2%) and HbA1c (mean change −0.3%), as well as step counts (mean change 869 steps/day). Overall, this intervention could be cost-prohibitive for under-resourced populations, such as Journey to Health participants, without sustainable funding and collaborative partnerships to support teaching kitchens at community sites.

Journey to Health participants also significantly increased fruit intake and decreased sugar sweetened beverage consumption. Daily fruit consumption increased by nearly half a cup (0.43 cups). Fruit, vegetable, and sugar sweetened beverage consumption were discussed throughout the course, but specifically highlighted in two of the three hands-on culinary demonstrations when participants learned how to make a strawberry and orange citrus salad (primarily fruit), as well as a sugar-free agua fresca, a popular Mexican drink that includes water and fresh fruit, for a suggested alternative to soda. These results coincide with those of other community nutrition interventions. Stauber et al. found participants were more likely to meet MedDiet points for fruit intake after a six-week hands-on community culinary education course [52]. Authors of that study posited that culinary education courses may offer a cost-effective approach to addressing nutrition related chronic diseases. Similarly, Sharma et al. found a significant increase in fruit and vegetable consumption amongst low-income participants with type 2 diabetes who attended the “Prescription for Healthy Living” culinary medicine program, a clinic-based intervention with five virtual classes and a nine-month food prescription [36]. Authors of that study emphasized that participants who face steep social determinants of health cannot simply be encouraged to increase fruit and vegetable consumption without experiencing a simultaneous increase in access to those foods, as this “reductionist philosophy” disregards financial, social, and environmental factors. Incentives provided to participants who attended the Journey to Health program were intended to address food insecurity (see Table 1); however, these were not sufficient to make a long-term or meaningful impact on the food access of participants. Therefore, future iterations of the Journey to Health program should explore incorporating food prescriptions or food pharmacy programs.

This Special Issue of Nutrients includes a focus on how health and wellness promotion strategies can be organized, evaluated, and optimized for impact. We used a process of integration of Journey to Health within existing community resources to ensure the intervention was practical and aligned with community priorities. The Wellness Bus includes community health workers and a registered dietitian with the training necessary to deliver Journey to Health. By integrating our initial development and evaluation with the Wellness Bus, we were able to design an intervention that was attractive to their clientele, but also included only four additional activities (nutrition classes and their corresponding incentives) to supplement the services already offered on the Wellness Bus, thus increasing the likelihood of both adoption and sustainability. Further, using pragmatic methods and existing services such as the biometric screens and health coaching appointments, the incremental costs of delivering Journey to Health are minimal, resulting in a higher potential for sustainability.

Applying the RE-AIM planning and evaluation model to community-based nutrition education and culinary skill training intervention development also provides a generalizable process to plan, initiate, evaluate, and sustain these interventions. Iterative planning approaches using RE-AIM have been successful in community and health care settings to address physical activity, healthy eating, and disease prevention [53,54,55,56,57]. Within Journey to Health, this approach included focusing primarily on the individual level and determining where the intervention should be delivered to increase access and on strategies to recruit participants (reach), and secondarily on the evidence-base to determine key intervention components (effectiveness). Considerations of adoption, implementation and sustainability also informed the location of intervention, the number of program activities that could be completed, the alignment of community priorities, and the potential for sustainability in terms of the Wellness Bus funding and personnel. While the presence of a Wellness Bus is not the generalizable part of our work, the RE-AIM planning process should be generalizable and support others interested in delivering Journey to Health to identify local place-based solutions for intervention development, delivery, and sustainability.

Strengths of our study include our success in incorporating essential elements of teaching kitchens including nutrition and physical activity education, culinary demonstrations, and the use of motivational interviewing strategies and health coaching techniques to encourage long-term behavior change. The incorporation of both didactic and experiential, or hands-on, elements is hypothesized to bolster both knowledge and skill acquisition. Hands-on participant activities included engaging in culinary demonstrations and tastings, interpreting and comparing nutrition facts labels using real examples of common foods, creating healthy meal plans using MyPlate as a guide, eating shared community meals that included program feedback discussions, and participating in biometric screens and health coaching appointments. A major strength of Journey to Health was the prioritization of health equity. Bilingual programming was offered in four historically under-resourced areas in Salt Lake County, with low rates of health insurance and high rates of chronic disease [58]. The culinary demonstrations in the fourth nutrition class were tailored to include ingredients typical of Latin American dietary patterns. Previous research has shown that culturally tailored culinary demonstrations have improved diet quality [59] and fruit and vegetable consumption [60]. Study participants who indicated social needs were referred to local resources by trained volunteers on the Wellness Bus [61]. Most importantly, preventative health services on the Wellness Bus remain free and available to participants post-intervention, improving healthcare access and the potential for sustained improvements in biometrics and health behaviors.

Another strength included the interprofessional collaboration amongst dietitians, community health workers, and Spanish language interpreters. Each member of the interprofessional implementation team was able to offer expertise in different areas, making it possible to offer biometric screens, health coaching appointments, and group education in both English and Spanish. Due to the varied format of program activities, participants were given the opportunity to learn in a group setting, then apply new knowledge to their own lives during individualized health coaching appointments. Elements of health coaching that have been identified as beneficial for health behavior change in past studies include individualization and the use of personalized strategies, perceived social support, accountability, and motivation [62,63]. This program also offered mentorship and community exposure to health professionals in training, including graduate level nutrition, health science, and medical students.

To note, there are several limitations of the study. First, the study was conducted in a single county with a specific under-served audience, and as such, the Journey to Health intervention and outcomes may not generalize more broadly. However, this limitation is balanced with the use of a well-established planning and evaluation process that could result in a generalizable impact, though using unique interventions tailored to a specific community. The study is also prone to self-selection bias since convenience sampling was employed, with the participants choosing to join a program cohort, instead of the study utilizing a randomized controlled design. Additionally, due to the voluntary nature of the study, participants were not required to attend any portion of the program. As such, only a limited proportion of the total participants met criteria for one of the analyses presented in this paper. Incentives provided at the nutrition classes and health coaching visits (see Table 1) may have encouraged attendance at these events; however, incentives were not provided for completing biometric screens, which may have influenced the number of participants who chose to complete the screens. Behavioral measures were self-reported, which can be prone to participant under or overreporting [64,65,66]. Yet, objective biometric data were also collected for this study, and the changes in health behaviors and objectively measured outcomes demonstrated similar changes. Also, due to the 6 month duration of the program, we are unable to observe long-term effects of the intervention on participant biometrics and health behaviors, such as with programs of one year duration or longer. To note, the largest limitation is the use of pre- and post-intervention evaluation without a comparison group, which removes our ability to understand if the changes were secular or related to intervention participation.

The next steps in our program of research include expanding testing of Journey to Health within a comparative effectiveness trial that will compare the full intervention to standard Wellness Bus services (i.e., biometric screenings and health coaching) on changes to participants randomly assigned to each condition. We also will continue to focus on strategies to integrate Journey to Health as a sustained offering by the Wellness Bus or other community-based organizations interested in implementation. Our informal follow-up at community meals indicated that participants have a significant interest in hands-on culinary education. In concert with a randomized trial, we will conduct focus groups with program participants who identify as Hispanic or Latino to obtain qualitative data that will guide the design process of new culturally tailored culinary medicine program content, structure, and follow-up activities to increase the likelihood of sustained health and health behavior improvements.

## 5. Conclusions

Overall, the findings of the Journey to Health pilot study provide preliminary support for the feasibility of nutrition education with culinary instruction and program delivery in community-based settings. In terms of practical applications to the area of community nutrition education, this study has demonstrated the importance of (1) utilizing a planning and evaluation framework, such as RE-AIM, (2) incorporating an intervention into pre-existing resources, (3) designing an intervention that fills existing resource gaps and considers accessibility, language, and cultural relevancy, and (4) including skill-building activities, utilized by teaching kitchens, such as culinary demonstrations. The rising trend of frequent fast-food consumption [67,68], paralleling the decline of home cooking and meal preparation [69], supports the need for nutrition education with culinary instruction in low-income and underrepresented minority populations, as low socioeconomic and underrepresented minority individuals report even less cooking of meals at home [69]. Thus, future research on innovative, community-based collaborative strategies are needed to demonstrate the impact of culturally appropriate nutrition education with culinary instruction on participant biomarkers, as well as health behaviors, and subsequent adoption, implementation, and sustainability outcomes at the community level. Lastly, further research is recommended to explore innovative and culturally tailored skill building strategies to address the barriers to home food preparation for individuals and families, in low-income and underrepresented minority populations.

## Figures and Tables

**Table 1 nutrients-16-00618-t001:** Journey to Health Program Overview.

Program Activity	Description	Incentive	Month
Registration	Call, text, or register online.	NA	0
Biometric Screen 1	Informed consent, then completion of the first biometric screen on the Wellness Bus.	NA	1
Nutrition Class 1	One-hour nutrition class. Topic: fruits and vegetables. Culinary demonstration: Strawberry and Orange Citrus Salad.	Sample of the culinary demonstration.	2 (week 1)
Nutrition Class 2	One-hour nutrition class. Topic: food assistance programs, and MyPlate.	Food box (fresh and shelf-stable foods).	2 (week 2)
Nutrition Class 3	One-hour nutrition class. Topic: nutrition fact labels and grocery shopping.	$25 grocery store gift card.	2 (week 3)
Nutrition Class 4	One-hour nutrition class. Topic: low-sodium and low-sugar snacks and drinks. Culinary demonstration: (1) sugar-free agua fresca, and (2) bean and corn salsa.	Sample of the culinary demonstration and one kitchenware item (e.g., spatula, whisk, citrus juicer).	2 (week 4)
Health Coaching Appointment 1	One-hour appointment with Wellness Bus registered dietitian who performs a health behaviors assessment, provides education, and facilitates goal setting. Sessions are tailored to meet the needs of the individual. Topics: dietary and physical activity habits, sleep duration, stressors, barriers to behavior change, and readiness to change.	$10 grocery store gift card.	3
Health Coaching Appointment 2	Thirty-minute follow-up appointment with Wellness Bus registered dietitian. Sessions are tailored to meet the needs of the individual. Topics: progression toward goals set in the previous appointment.	$10 grocery store gift card.	4
Biometric Screen 2	Second biometric screen on the Wellness Bus.	NA	5
Community Meal	Participants share a meal and provide feedback on the program.	Meal.	6

**Table 2 nutrients-16-00618-t002:** Self-Reported Health Behavior Questions.

Health Behavior Measure	How It Was Assessed
Physical Activity	On average, how many days per week do you accumulate at least 30 min of moderate to high-intensity physical activity like brisk walking, housework, cycling, swimming, or sports?
Fruit Intake	On average, how many total cups of fruits do you eat each day (e.g., 1 cup sliced or chopped fruit or 1 piece of softball-sized fruit)?
Vegetable Intake	On average, how many total cups of vegetables do you eat each day (e.g., 1 cup sliced or chopped vegetables; if raw, leafy greens then count 2 cups as 1 cup)?
Sugar Sweetened Beverage Intake	On average, how many sugar sweetened beverages do you drink in a day? Sugar sweetened beverages include regular soda, juice, lemonade, chocolate milk, energy or sports drinks coffee drinks such as mochas, lattes.
Artificially Sweetened Beverage Intake	On average, how many artificially sweet drinks do you drink in a day? Artificially sweetened beverages include diet soda, light juice, low calorie energy or sports drinks.
Dietary Change Self-Efficacy	At the present time, how sure are you that you can make and stay with changes in your diet? (self-efficacy)

**Table 3 nutrients-16-00618-t003:** Program Activities and Corresponding Time Parameters to Meet Eligibility for Analysis.

Program Activities	Time Parameters
Biometric Screen 1	Completed either three months prior to the first nutrition class, or up to one month following the first nutrition class.
Health Coaching Appointment 1	Completed either three months prior to the first nutrition class, or up to the date of the community meal.
Health Coaching Appointment 2	Any secondary appointment completed after the initial appointment, but no more than three months after the community meal.
Biometric Screen 2	Any secondary biometric screening completed after the initial screen, but no more than three months after the community meal.

**Table 4 nutrients-16-00618-t004:** Participant Characteristics.

Age- *Mean* (*SD*)	51.45 (15.31)
Gender- *n* (%)	
Female	99 (79.2%)
Male	25 (20%)
Other	1 (0.8%)
Race- *n* (%)	
American Indian	2 (2.15%)
Asian	11 (11.83%)
Black or African	3 (3.23%)
Native Hawaiian or Pacific Islander	2 (2.15%)
White	36 (38.71%)
Other	39 (41.94%)
Ethnicity- *n* (%)	
Hispanic or Latino	76 (61.79%)
Not Hispanic or Latino	47 (38.21%)
Language- *n* (%)	
English	76 (58.91%)
Spanish	53 (41.09%)
Nutrition Classes Attended- *Mean* (*SD*)	3.02 (1.06)
Biometric Screenings- *Mean* (*SD*)	3.05 (4.50)
Health Coaching Appointments- *Mean* (*SD*)	2.03 (1.39)
Days Between Biometric Screens- *Mean* (*SD*)	133.19 (62.15)
Days Between Health Coaching Visits- *Mean* (*SD*)	55.07 (32.84)

**Table 5 nutrients-16-00618-t005:** Changes in health outcomes for those with at least two biometric screenings and one nutrition class attended.

Measure	Initial Screen	Follow-Up Screen	Mean Change (95% CI)	*p*-Value
BMI (*n* = 91)	29.22 (5.89)	28.96 (5.71)	−0.25 (−0.49, −0.03)	0.029
Systolic Blood Pressure (*n*= 107)	125.17 (17.97)	120.96 (15.21)	−4.06 (−7.12, −1.29)	0.005
Diastolic Blood Pressure (*n* = 107)	80.59 (9.85)	77.38 (8.94)	−3.21 (−4.84, −1.57)	< 0.001
Non-Fasting Glucose (*n* =70)	123.53 (57.82)	116.70 (40.21)	−6.83 (−17.41, 3.76)	0.203
HbA1c (*n* = 65)	6.09 (1.21)	6.01 (1.01)	−0.08 (−0.25, 0.08)	0.324
Total Cholesterol (*n* = 64)	179.33 (38.43)	175.03 (35.85)	−4.30 (−14.56, 5.96)	0.406
HDL (*n* = 68)	49.38 (14.04)	49.18 (13.39)	−0.21 (−2.85, 2.44)	0.877
LDL (*n* = 51)	92.90 (34.98)	96.61 (31.37)	3.71 (−7.32, 14.73)	0.503
Triglycerides (*n* = 62)	216.61 (109.88)	163.1 (143.53)	−53.52 (−80.67, −26.36)	< 0.001

**Table 6 nutrients-16-00618-t006:** Changes in health outcomes for those with a high-risk initial measurement and at least two biometric screens and one nutrition class attended.

Measure	Initial Screen	Follow-Up Screen	Mean Change (95% CI)	*p*-Value
BMI ≥ 25 (*n* = 67)	31.55 (5.04)	31.24 (4.83)	−0.31 (−0.60, −0.12)	0.038
Systolic Blood Pressure ≥ 120 (*n*= 65)	135.51 (14.80)	127.29 (14.28)	−8.22 (−12.31, −4.12)	<0.001
Diastolic Blood Pressure ≥ 80 (*n* = 61)	87.13 (6.36)	81.07 (8.26)	−6.07 (−8.06, −4.07)	<0.001
Non-Fasting Glucose ≥ 200 (*n* = 4)	323.25 (100.10)	196.75 (93.91)	−126.50 (−269.50, 16.50)	0.067
HbA1c ≥ 5.7 (*n* = 30)	6.82 (1.46)	6.58 (1.24)	−0.24 (−0.59, 0.12)	0.185
Total Cholesterol ≥ 200 (*n* = 18)	226.89 (16.48)	192.67 (30.82)	−34.22 (−53.08, 15.36)	0.001
HDL ≤ 40 (*n* = 22)	35.77 (3.78)	38.68 (8.66)	2.91 (−0.97, 6.78)	0.133
LDL ≥ 100 (*n* = 21)	124.10 (23.22)	105.76 (29.77)	−18.33 (−35.10, −1.57)	0.034
Triglycerides ≥ 150 (*n* = 42)	267.21 (96.52)	188.43 (73.91)	−78.79 (−114.77, −42.80)	<0.001

**Table 7 nutrients-16-00618-t007:** Changes in health behaviors for those with at least two health coaching appointments and one nutrition class attended.

Measure	Initial HC Appt.	Follow-Up HC Appt.	Mean Change (95% CI)	*p*-Value
Days with 30 Minutes of MVPA Per Week (*n* = 72)	2.72 (1.25)	2.82 (1.20)	0.10 (−0.15, 0.35)	0.441
Cups of Fruit Per Day (*n* = 68)	2.01 (0.97)	2.44 (0.94)	0.43 (0.21, 0.65)	<0.001
Cups of Vegetables Per Day (*n* = 68)	2.18 (0.93)	2.32 (1.03)	0.15 (−0.06, 0.35)	0.159
Number of Sugar Sweetened Beverages Per Day (*n* = 56)	1.21 (0.53)	1.02 (0.13)	−0.20 (−0.34, −0.49)	0.010
Number of Artificially Sweetened Beverages Per Day (*n* = 55)	1.16 (0.57)	1.05 (0.40)	−0.11 (−0.25, 0.04)	0.134
Diet Self-Efficacy (*n* = 24)	3.71 (0.55)	3.42 (0.83)	−0.29 (−0.65, 0.07)	0.110

Note. MVPA = Moderate to vigorous physical activity.

## Data Availability

Data available upon reasonable request.

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
