# Peer review of "Evaluation of the Effectiveness of a Bilingual Nutrition Education Program in Partnership with a Mobile Health Unit"

_nutrients, 2024, doi:10.3390/nu16050618_

Round 1

Reviewer 1 Report

Comments and Suggestions for Authors

1)It is recommended that the author strengthen the motivation for this study, including an examination of the current status of nutrition education and cooking teaching in the experimental area, and provide a clear explanation of the study's significance.

2)There is a lack of analysis of previous literature on community nutrition education.

3)The sampling method and theory are not adequately addressed. Is the sample representative? There is no specific explanation of the reliability and validity of the Self-Reported Health Behavior Questions used.

4)There is also a deficiency in the analysis and explanation of the background information of the sample. For example, it is unclear whether factors such as gender and nationality might contribute to differences in nutritional knowledge due to previous family influences.

5)The study lacks a discussion on research limitations and a description of practical contributions to nutrition education.

Reviewer 2 Report

Comments and Suggestions for Authors

Dear Authors,

The manuscript (nutrients-2825516) submitted for review is very interesting.  It points the possibility of using culinary classes to promote health. The study showed an improvement in indicators such as body mass index, blood pressure, and triglycerides. However, the results of the study may be accidental, as the presented strategy was not long-term. Nevertheless, the results obtained indicate the sense of using strategies such as The Journey to Health. The issue of culinary classes and learning to prepare meals yourself is very important because, firstly, it limits the consumption of food in fast food establishments, especially when consumers have low incomes. Moreover, it also teaches good food raw materials choices for their preparation.

Technical comments

In my opinion, the authors should have separated two sections in the Limitation and Conclusions article.

 The conclusions need to be improved and should direct these or other scientists to further research.

References: References are cited according to journal rules.

Reviewer

Round 2

Reviewer 1 Report

Comments and Suggestions for Authors

Thank you for your response and correction.

Author Response

Thank you for taking the time to review our manuscript, we appreciate your feedback and believe your comments have strengthened our paper.